# How are people coping with working from home during the COVID-19 pandemic?: Experiences from the Netherlands and South Korea

**So Yeon Park**[1,2]*, **Rachel Lee**[1], **Caroline Newton**[1], **Gisung Han**[3]

**1** Department of Urbanism, Faculty of Architecture and the Built Environment, Delft University of Technology (TU Delft), Delft, The Netherlands, **2** School of Architecture, Seoul National University of Science & Technology, Seoul, Republic of Korea, **3** Institute of Engineering Research, Korea University, Seoul, Republic of Korea

* s.y.park@seoultech.ac.kr

**Data Availability Statement:** All relevant data are within the paper and its Supporting Information files.

## Abstract

COVID-19 has made working from home routine for many. People who have had to maintain their productivity, particularly in physically and/or socially unacceptable home-working situations, experienced one of the pandemic's disadvantages. The experience can vary substantially among individuals as well as by country. This study presents the results of a comparative study of the Netherlands and Korea. Working from home was not uncommon in the Netherlands before the pandemic; however, in Korea, employers adopted working from home from its start, and that increased rapidly. An online survey enabled us to compare the physical and social conditions of current home workspaces in both countries, to understand how well-equipped they were to support people who had to work from home. We studied the changes in productivity and physical/mental health before and during COVID-19, to learn how people coped with working from home in both countries. Contrary to expectations, Koreans showed better scores than people in the Netherlands, in terms of changes in health and productivity. This article discusses various aspects of that result, such as satisfaction with home workspace, housing type, job position and prior experience, compulsoriness, and frequency of working from home. Relieving stress and concentration appeared to be the most important dimensions of telecommuters' satisfaction with working from home environments in both countries. The results are the basis for suggesting the development of strategies for a desirable WFH environment, considering different background contexts, experiences and cultures.

## 1. Introduction

The coronavirus disease identified in 2019 (COVID-19) generated a crisis that has made working from home (WFH) a routine for many. Prior to the pandemic, WFH had steadily increased throughout the preceding decade. Only 5.4% of employed people in the EU-27 worked from

**Funding:** This study was supported by the Research Program funded by the SeoulTech(Seoul National University of Science and Technology).

**Competing interests:** The authors have declared that no competing interests exist.

home on a regular basis, a ratio that has remained relatively stable from 2009 until the onset of COVID-19. Additionally, only 15% of EU workers had ever worked from home prior to the pandemic [1]. However, a survey by Eurofound [2] concluded that 40% of employees in the EU started WFH full-time as a consequence of COVID-19. The United States showed similar figures. Only 3% of full-time workers in the USA primarily worked from home in 2017, and the proportion of those who telecommuted more than four days per month also remained around 10% in the same year [3]. However, according to the U.S. census in 2020, 36.9% of survey participants answered that at least one adult in their household (including themselves) had replaced some or all of their face-to-face work with WFH during the pandemic [4].

Catalysed by COVID-19, this sudden and unpredictable expansion of WFH was global and general. It took place across the world, including contexts where WFH had not previously been widely implemented, and those involved were largely unaware of possible issues. Its uneven impact has led people who must maintain their work productivity, particularly in physically and/or socially unacceptable home-working situations, to experience one of the pandemic's disadvantages. The experience can vary substantially among individuals as well as countries.

This study presents the results of a comparative survey-based study of the Netherlands (NL) and South Korea (KR). The two countries share high scores on the Information and Communications Technology (ICT) Development Index, a prerequisite for successful WFH. As of 2017, KR ranked second, and NL ranked seventh on the Index [5]. However, they differ in various aspects of WFH.

WFH was not uncommon in NL prior to the pandemic. About 14.1% of the Dutch workers worked from home in 2019, when the European average was around 5% [6, 7]. BBC News recently noted that 'the Netherlands may have figured out something about working from home (pandemic or no) that the rest of the world has yet to learn' [8]. The number of people working from home (about 40% of Dutch residents) has increased even more since March 2020, when the Dutch government strongly advised employees to work from home as much as possible to contain the spread of COVID-19 [9]. However, in South Korea, where WFH was unusual before COVID-19, it increased rapidly from the start of the pandemic. One study performed a shift-share analysis, based on the rates of WFH in 2000 and 2010, and estimated that the rate of WFH in 2020 would reach 1.33% in the metropolitan area and 1.23% nationwide, which shows that WFH in KR was not expected to grow as much as it had in Europe or the USA [10]. Yet, according to the results of a survey that Gallup Korea [11] conducted of 1,204 office workers aged 25–54, 30% of respondents had worked from home in the past year during the pandemic, and 25% of them had experienced WFH for the first time since COVID-19. In addition, in 2020, the number of telecommuting workers in public institutions in KR appeared to be 162,618, 115.5 times the number in 2019 [12]. These statistics imply that WFH, a substantial measure to prevent the spread of coronavirus, would have been a more unfamiliar and unanticipated form of work for Koreans than for people in NL.

WFH-related measures are another important factor that may have influenced telecommuters' experience during COVID-19. The results section discusses further the WFH measures the two countries took during the pandemic, which differed in their intensity and use of coercion. During the pandemic, WFH measures had a direct impact on the compulsory nature, duration and frequency of WFH, which strongly affected telecommuters' emotional, perceptual or behavioural responses [13].

In addition to WFH-related factors, the cultural differences between the two countries must be a criterion for determining their respective suitability for WFH [14]. Himawan in 2022 [15] defined the cultural characteristics of Asian countries using the Hofstede model of cultural dimensions [16], to suggest the sociocultural barriers that hinder WFH in these

nations. The cultural profile of Asian nations that the authors presented (i.e. high levels of power distance, masculinity, collectivism, uncertainty avoidance and low-level indulgence) appears applicable to KR as well, according to Hofstede's insights [17]. They suggest that these cultural dimensions would result in possible barriers to WFH, such as lack of supervisory control, an unsatisfied sense of prestige, social disconnection and isolation, resistance to adopting new ways of working and working overtime, due to feelings of guilt associated with increased autonomy. Conversely, NL has very different cultural characteristics from KR's. NL is a country with low-level power distance and masculinity, high-level individualism and indulgence according to Hofstede's insights [17]. On the basis of these national cultural differences, we attempted to address the differences between the two countries' work cultures, in the context of telecommuters' experience in each.

The following are the hypotheses of the present study:

H1: Changes in physical/mental health and productivity among telecommuters in KR and NL during COVID-19 may have been affected by different predictors.

H2: There will be differences in the physical/mental health and productivity changes experienced by telecommuters in KR and NL during COVID-19.

H3: Among the variables that comprise satisfaction with the WFH environment, predictors of changes in physical/mental health and productivity of telecommuters during COVID-19 may differ between KR and NL.

In addition, based on the results of testing H1 described above, the following hypotheses were generated regarding predictors that showed interesting differences between the two countries: There will be differences in physical/mental health and productivity changes between those living in apartments and houses (H4); between those with and without prior WFH experience (H5); according to the frequency of WFH and its compulsoriness (H6); and depending on the amount of interaction required during work (H7).

Now that the global COVID-19 pandemic has officially ended, some major employers are urging employees to return to the office more frequently, but many employees have come to appreciate the advantages of remote work. Experts recommend considering remote and hybrid arrangements as viable long-term alternatives [18]. Our data support this suggestion, showing that 77.2% of participants preferred a mixture to working fully from home (10.9%) or in the office (11.4%). This study sought to investigate and compare the WFH experiences of individuals in KR and NL. By examining the factors that influence these experiences, we aim to provide valuable insights that can inform the perpetuation of WFH policy. We expect to contribute to the development of evidence-based recommendations that can enhance the quality of remote work environments and the well-being and productivity of telecommuters.

## 2. Materials and methods

### 2.1. Study design and sample

To collect data for this study, we created an online survey questionnaire in Qualtrics. The Human Research Ethics Committee at TU Delft reviewed and approved the design and distribution of the questionnaire and the data management plan on November 1, 2021. The online survey was open from November 4 to December 29, 2021, and distributed through various online networks, including Facebook, LinkedIn and KakaoTalk, which is a dominant social networking service in KR, to randomly recruit participants. Participants were limited to those who worked from home during the pandemic in NL or KR, who had worked in a space other than home before COVID-19, and who performed their work at a desk for a significant

amount of time. We also excluded Koreans living in NL because we recognized that they may have been influenced by the culture and policies of both KR and NL, which can affect the consistency and interpretability of our findings. Written informed consent was provided in the online survey before participants began the survey, and all participants agreed to the survey process and the data management plan for this study. In accordance with our data management plan, no information that could identify individual participants was accessed during or after data collection. The questionnaire comprised questions designed to measure all the variables described in Section 2.2.

A total of 624 people accessed the survey link to start the questionnaire, but we classified 315 people as unsuitable for participating, using the following questions: (1) During COVID-19, have you ever worked from home?, (2) Before COVID-19, did you only work from home? (3) Does your work involve sitting at a desk? (4) (in the NL questionnaire) Are you Korean? If a participant answered 'no' to questions (1) or (3) or 'yes' to questions (2) or (4), the survey did not proceed further. Among the 309 completed responses, we excluded two due to insincerity, bringing the total number of participants to 307: n = 195 respondents in KR and n = 112 in NL. Among participants in NL, most were Dutch (58.0%), 6.3% were German, 4.5% were Italian, 3.6% were British, 2.7% were American, 2.7% were Greek, 1.8% were Indian and the rest were from other countries in Europe. There were more female participants (n = 196) than male (n = 109), and two participants preferred not to reveal gender. The majority (46.6%) of participants were aged 31 to 40. Five participants were aged under 21, and nine were 51 years or older. Married participants represented 45.6% of participants, with another 6.8% in a domestic partnership, and 42.3% were single while 3.0% were widowed, divorced or separated. Seven participants preferred not to declare marital status.

## 2.2. Research variables

**2.2.1. Outcome variables.**   In a model developed by Vischer [19] to provide dimensions for designing a functionally comfortable workplace, workers' 1) productivity and 2) physical, 3) mental and 4) social health were presented as achievable objectives. We adopted these as our outcome variables, to objectively quantify various aspects of the telecommuter's experience during the pandemic. Additionally, we have included the 5) work-life balance variable in the set of outcome variables, as this is an issue of particular importance for telecommuters.

Among the five outcome variables, physical/mental health and productivity were the subject of several subquestions on our survey. Questions on overall physical health, 24-hour cycle and drowsiness while working also appeared among the subquestions. Also, in addition to general mental health, the survey included questions regarding sleep quality, depression and work-related stress. Referring to the questionnaire that Pitchforth in 2020 [20] developed, we requested participants' levels of job satisfaction, work engagement, work enjoyment and energy and concentration while working.

To determine how the introduction of WFH during the pandemic has *changed* workers' well-being, we measured differences between periods before and during the pandemic, using a 5-point Likert scale for each outcome variable. For example, two questions about physical health were asked: 1) In general, how would you rate your physical health when you were not working from home before COVID-19? and 2) In general, how would you rate your physical health when you were working from home during COVID-19? (1: terrible, 2: poor, 3: average, 4: good, and 5: excellent). To calculate changes in health and productivity, we subtracted the 'before' score from the 'after' score. Thus, negative values for these variables indicated a deterioration in health or productivity during COVID-19, while positive values meant improved health or productivity.

**2.2.2. Predictive variables.** As the first variable to predict the health and productivity changes of telecommuters during the pandemic, we considered the quality of their home workspace. Several researchers have investigated the effect of home office features on occupants' productivity during WFH. Xiao in 2021 [21] investigated how social, behavioural and physical factors affected office workstation users' well-being while working from home. They included the visual/thermal environment, air quality and noise, as well as the presence of a dedicated space for work and a well-designed workstation, as elements of a home office environment. They discovered significant relationships between these elements and physical/mental health. Pang in 2021 [22] exclusively focused on housing's indoor built environment for insights into how the design of indoor space improves WFH operation by employing Indoor Environmental Quality (IEQ) measurements. These indicate the quality of the built environment as the occupants experience it. Measuring objective indoor factors, such as lighting, acoustics, thermal environment and air quality, can contribute to evaluating IEQ.

On the other hand, some researchers have focused on the importance of the occupant's *perception* of IEQ instead of IEQ itself. Pang in 2021 [22] revealed a strong correlation between telecommuters' ratings for IEQ of their WFH environment and their physical/mental health and productivity. Likewise, Chen in 2020 [23] demonstrated that IEQ satisfaction is the strongest predictor of occupants' work productivity, particularly in private offices. In this regard, we used telecommuters' ratings of their home workspace as criteria for evaluating the current conditions of home workspace and predictors of changes in health and productivity during COVID-19. In addition to general IEQ factors, such as satisfaction with temperature, natural lighting, noise, aesthetical pleasure, comfort and privacy, we measured how well telecommuters could concentrate and de-stress while performing WFH in their home workspace. Finally, we added 'attachment to home', referencing the findings of [24] on the significant relationships between attachments to home and residents' mental health during the COVID-19 pandemic.

In addition to the telecommuter's evaluation of the physical work environment, we considered the social aspects of the home-based work environment, including whether the home included cohabitants and children. As for the social environment outside the home, we considered participants' attachment to their neighbours and whether they lived in an apartment building or a house.

We included another predictor that related to the implementation of WFH. As mentioned previously, KR and NL had different prior experiences with WFH and adopted slightly different positions regarding WFH measures during the pandemic. We assumed that preparedness for WFH and attitudes that governments or companies took towards WFH may have influenced telecommuters' experience. To account for this, we included as predictors the participant's prior experience with WFH before COVID-19, the frequency of WFH per week and the mandatory nature of WFH.

Included among the demographic variables were nationality, country of residence, age, sex, marital status and occupation-related factors. Numerous recent studies have demonstrated the significant effect of job insecurity or financial worries on anxiety and depression among workers during the COVID-19 pandemic—e.g. [25, 26]. To include job insecurity as a predictor variable, we investigated whether they were full-time employees, part-time employees, self-employed or interns. To incorporate the different workplace cultures of the two countries into occupation-related variables, we added two other variables, i.e. job position and the perceived level of interaction with colleagues or superiors *required* at work. One study classified a culturally diverse group of workers using two of Hofstede's dimensions: power distance and individualism/collectivism [27]. We also considered these two dimensions because NL and KR have significant cultural differences, particularly in power distance and individualism/collectivism

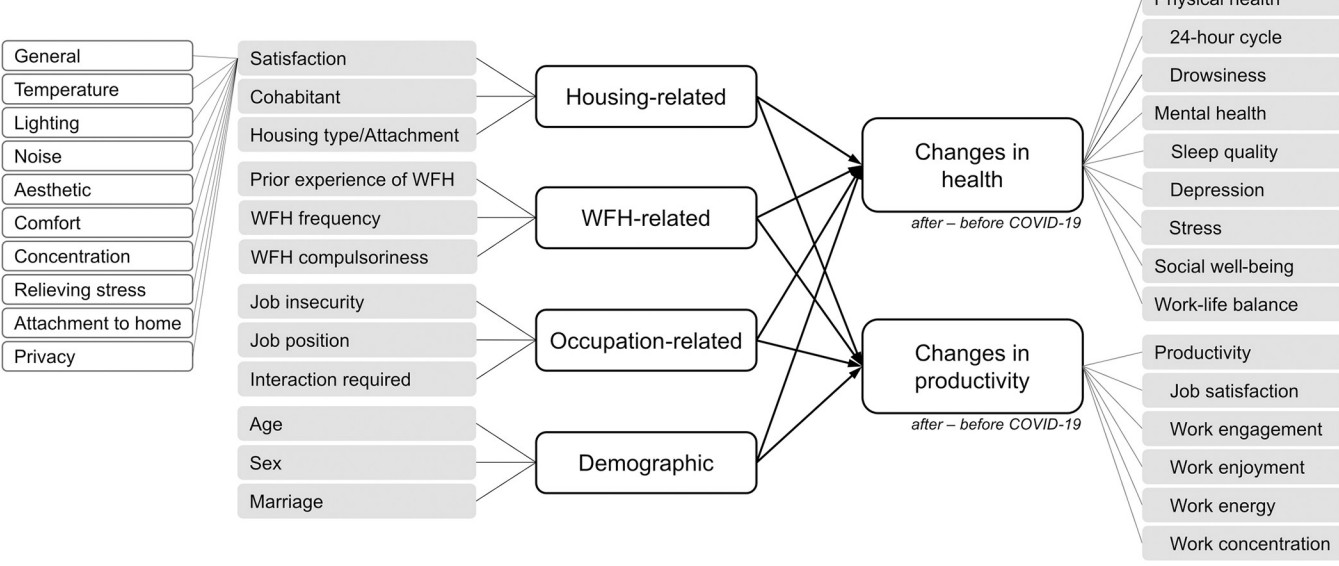

**Fig 1. Research variables and framework of this study.**

[17]. Regarding power distance, we hypothesised that the workers' power (i.e. job position) is likely to influence his or her WFH satisfaction, particularly in a work culture with high power distance. Moreover, we expected that the degree of interaction required with coworkers would impact telecommuters in a collectivist work culture more than those in an individualistic work culture. Fig 1 shows this study's variables and research framework, and S1 Appendix describes how these variables were measured and coded.

## 2.3. Statistical analyses

The statistical analyses were conducted in the IBM SPSS 28.0. Stepwise multiple linear regression analyses (alpha to enter = 0.05; alpha to remove = 0.1) were performed for each of the KR and NL respondents with physical/mental health and productivity changes as dependent variables to test our first hypothesis. All regressions were checked for uncorrelatedness of residuals, multicollinearity, homoscedasticity, normality, and linearity. The remaining hypotheses (H2-7) were tested using independent-samples t-tests (two-tailed), with data grouped according to the assumption of each hypothesis. A significance level of 0.05 was applied for all statistical analyses.

## 3. Results and discussion

Stepwise multiple regression analyses were performed for KR and NL, respectively, to examine the impact of predictors on changes in telecommuters' health and productivity (H1). All regression models were statistically significant. Fig 2 presents the $R^2$ of each model as well as the standardised regression coefficients for the predictors that had a statistically significant effect on each outcome variable. In addition to the regressions, an independent-samples t-test was conducted to compare KR with NL, in terms of all outcome variables (H2). The mean values of each outcome variable and the t-test results appear in Fig 2. Full results of regression analyses and the t-test can be found in the S2 Appendix and S3.1 Table in S3 Appendix, respectively.

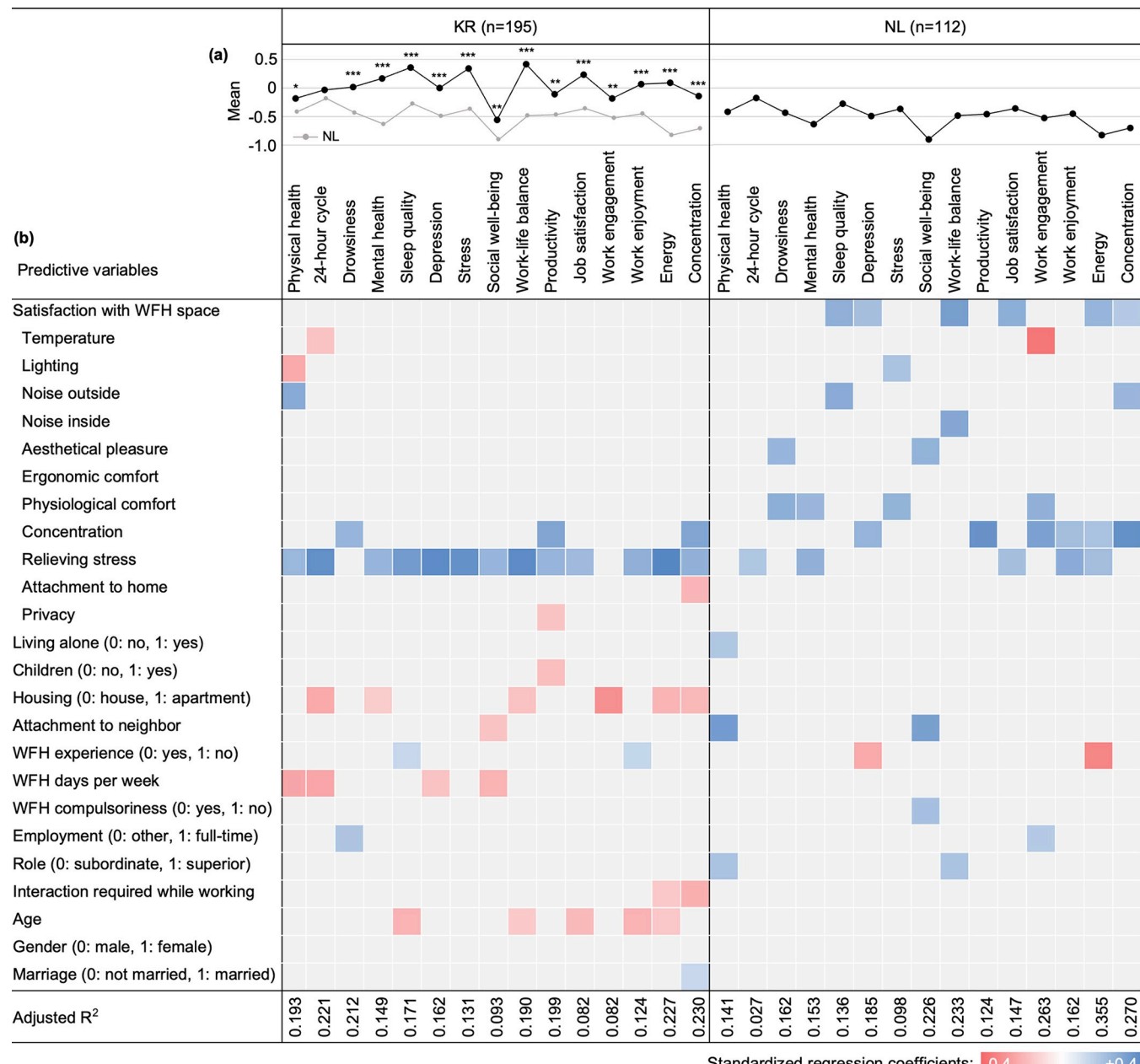

**Fig 2.** (a) Comparison between KR and NL on changes in telecommuters' health and productivity during the pandemic; (b) Regression models for outcome variables. Outcome variables with significant t-test results are marked with asterisks in the line graph (*$p<0.05$, **$p<0.01$, ***$p<0.001$). In the table below, predictors not included in the final regression models are shaded in grey.

Comparing KR and NL revealed statistically significant differences for all variables except for the 24-hour cycle. Interestingly, the mean values of KR respondents were greater than those of NL participants for all variables. Specifically, KR even showed improvements in work-life balance, stress, sleep quality, job satisfaction and mental health when working from home during COVID-19, while NL only showed negative mean values. This is somewhat surprising, as we expected NL to have been more socially, organisationally and individually prepared for WFH than KR, and that this readiness would have made NL participants better able to adapt to the sudden implementation of WFH.

The regression analysis results also confirmed differences between the two countries. In the case of KR respondents, satisfaction with the WFH environment in terms of relieving stress turned out to be the only variable to have had a strong influence on most health and productivity outcome variables. Conversely, for NL participants, different aspects of satisfaction with the home workspace affected outcome variables in various ways. Distinguished patterns also appeared for other predictors. For instance, past experience with WFH prior to COVID-19 showed contrasting effects in each country. Unlike KR, where having no prior experience had a positive impact on sleep quality and work engagement, we found that having had no previous experience negatively affected depression and work energy among NL participants. Also, only KR respondents confirmed significant negative impacts on health and/or productivity outcomes by living with children, WFH frequency, interaction demand while working and age. Additionally, whereas housing type had a substantial effect on outcome variables for KR participants, no such significant result appeared for NL participants. On the other hand, attachment to neighbours had a strong favourable influence on the physical health and social well-being of NL participants only, while negatively influencing KR participants' social well-being. Also contrary to our expectations, job position (whether subordinate or superior) significantly impacted only the NL respondents' physical health and work-life balance.

## 3.1. Satisfaction with the WFH environment (H3)

The most notable difference between the two countries regarding satisfaction with the WFH environment (shown in Fig 2) was the diversity of predictors that had statistically significant effects on the outcome variables. In KR, home workspace satisfaction, in terms of stress relief, had a very strong and exclusive impact on the majority of outcome variables. In NL, all predictors except ergonomic comfort showed varying impacts on the outcome variables. Nonetheless, we also discovered a commonality in both KR and NL, namely, satisfaction, in terms of stress relief and concentration, appeared to be important predictors for telecommuters' health and productivity. In many previous studies, occupants' subjective assessment of their living environments relied primarily on IEQ indicators (i.e. acoustical and thermal comfort, lighting and air quality), safety and privacy—e.g. [28–30]. However, our results suggest that for evaluating houses that accommodate both living and working functions, how the living environment can help residents relieve *stress* and improve *concentration* should be considered as important as comfort, safety and privacy.

The result showed an interesting finding for 'attachment to home', namely, it had a negative effect on the work concentration of KR respondents but no significant effect on NL participants. Given the many studies that have proved the beneficial impact of attachment to home on residents' well-being—e.g. [24, 31], these findings seem somewhat surprising. Yet, the findings of previous studies that stressed the significance of separating work and family to boost WFH productivity may provide a hint for interpreting these results. In research examining how migrants conceptualise 'home', Bivand in 2014 [32] emphasised the great importance of families in determining perceptions of 'home'. Considering the conceivable positive association between family ties and home attachments, we can assume that this attachment may have a detrimental effect on a telecommuter's productivity and concentration while working. For the home environment to be productive and efficient as well as emotionally comfortable and safe, the home should allow its occupants to take control of their physical, psychological and emotional connections to their home or family, according to their needs and wants.

## 3.2. Housing type (H4)

In Fig 2, the regression analysis indicates that for the KR participants, the type of housing—whether an apartment or a house—significantly influences certain outcome variables.

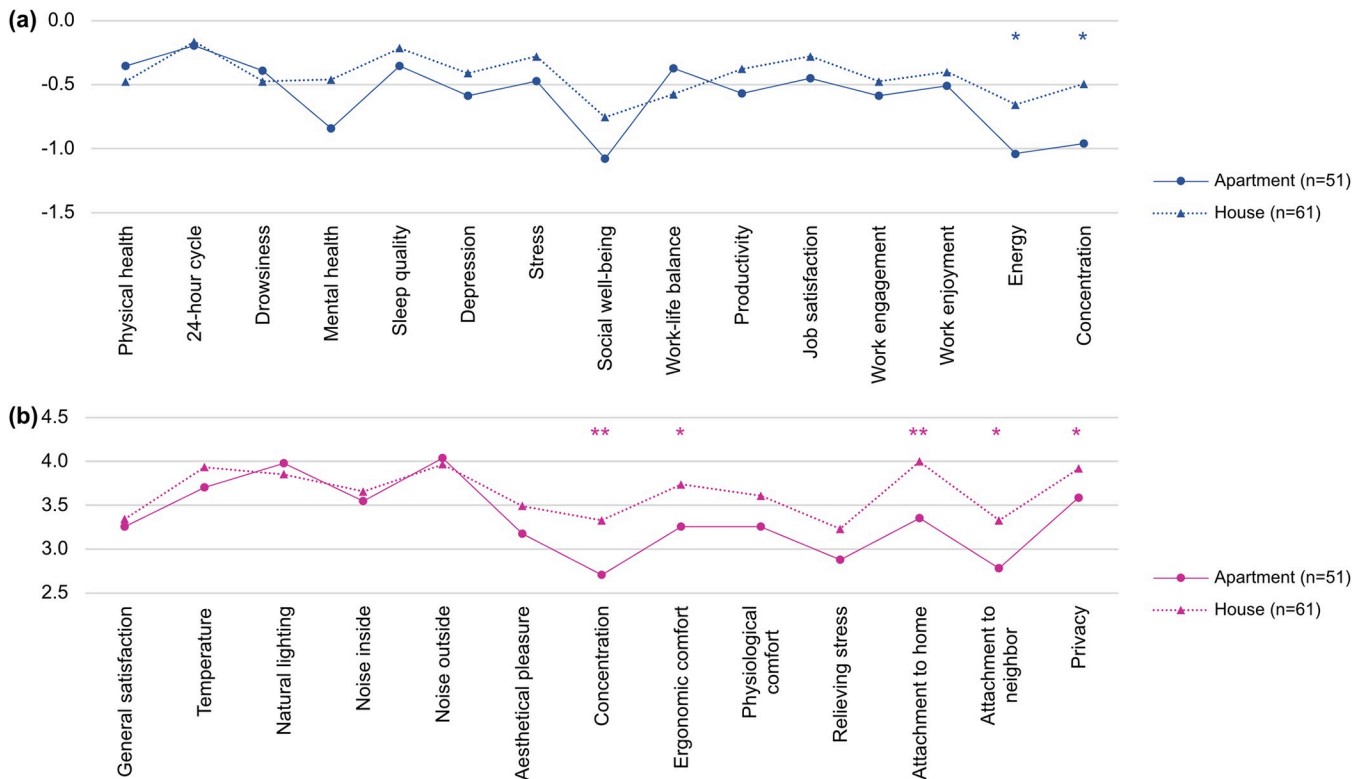

**Fig 3.** Comparison between people living in apartments and houses for NL participants on (a) changes in health and productivity; (b) satisfaction with home workspace. *$p<0.05$, **$p<0.01$, ***$p<0.001$.

However, this finding might not be fully representative, given that a large proportion (95.8%) of the KR participants resided in apartments, multifamily houses, or studio apartments. Therefore, to gain a broader understanding of the potential impact of housing type on WFH, we analysed data from the NL participants. We divided them into an apartment group (living in apartments, dormitories or studio apartments) and a house group (living in detached, semidetached, terraced houses or townhouses) to compare apartments and houses. For this comparison, we included as test variables not only changes in the health and productivity of telecommuters but also subvariables of satisfaction with 'WFH environment' presented in Fig 1. The results of the t-tests are shown in Fig 3, and the full results can be found in the S3.2 and S3.3 Tables of S3 Appendix.

Participants living in houses experienced significantly better energy ($t = 2.062$, $p = 0.042$) and work concentration ($t = 2.299$, $p = 0.023$) than those living in apartments. There were also significant differences between these two groups regarding satisfaction with their home workspaces, in terms of concentration ($t = 2.865$, $p = 0.005$), ergonomic comfort ($t = 2.466$, $p = 0.015$), attachment to home ($t = 3.493$, $p = 0.001$) and neighbours ($t = 2.421$, $p = 0.017$) and privacy ($t = 2.043$, $p = 0.044$). The house group had higher mean scores than the apartment group for all variables except for physical health, drowsiness, work-life balance and satisfaction with natural lighting and noise outside, although the differences were not statistically significant.

Interestingly, even though people living in apartments share a building with their neighbours, the average score of attachment to neighbours was significantly lower in apartments than in houses. Although explaining how housing type affects the residents' perception of their

homes or neighbours would require further research on this issue, the result of this study may hint that people living in apartments tend to have an emotional disconnection or sense of distance from their neighbours or surroundings, despite less physical distance from their neighbours than that of those living in houses, as a recent Korean study pointed out [33].

Attachment to neighbours also showed an interesting result in the regression analysis in Fig 2. Only NL respondents demonstrated a significant positive impact of this attachment on several outcome variables, particularly physical health and social well-being. Unlike NL, in the Korean group, the attachment to neighbours was found to have a rather negative effect on social well-being. To understand this difference between the two groups, we paid attention to the type of housing in which the participants lived. Whereas NL participants lived in various housing types, most Korean participants (72.8%) lived in apartment complexes, which can be defined as a group of high-rise buildings with community and commercial facilities within a residential complex, and the rest mostly lived in multifamily houses (17.9%), which are single three- or four-story residential buildings with in which two to nineteen households reside together, or studio apartments (5.1%). This issue of uniformity of housing type and negative aspects of the dominant housing type in KR has been discussed for a long time [34, 35]. Moon in 2020 [33] pointed out that the issue of the lack of communication between neighbours in apartment complexes in KR, which emerged in the 1970s and has been intensifying, is becoming more relevant in current times. In this sense, we assume that Koreans who have lived in apartments most of the time may not have high expectations for attachment to neighbours in the first place. Even though this issue requires further discussion from various points of view, in this study, we raised another research question regarding the type of housing and its impact on WFH.

### 3.3. Prior experience with WFH (H5)

Fig 2 showed that having no prior WFH experience positively affected KR participants and negatively affected those in NL. Our initial assumption was that previous experience with WFH would positively influence people who had to work from home during the pandemic, by having adequately prepared them physically, emotionally and/or psychologically. However, this assumption applied only to NL, not to KR. These conflicting results must be considered together with the social background of the two countries. At the time WFH was first recommended during COVID-19 in KR, it could have been a somewhat unfamiliar and new form of work to most Koreans. On the other hand, those in NL might have had direct or indirect experience with various types of remote work. In our survey, only 21% of KR respondents had experienced WFH prior to the pandemic, while nearly half (42.9%) of NL respondents had.

In this regard, we hypothesised that the lack of prior experience with WFH would have a rather positive effect on WFH novices in KR, where this type of work was newly introduced. This idea can be underpinned by borrowing the concept of the U-curve model of culture shock, which explains how immigrants adapt to a new culture in stages (see S4 Appendix). Culture shock happens when a person moves from familiar surroundings to an unfamiliar context [36]. In this sense, it might be reasonable to consider the change from working at an office to working from home as a big cultural transition in terms of work.

According to the U-curve model, the initial phase is the 'honeymoon' stage, when people enter a new culture and enjoy the excitement and curiosity. As Lysgaard [37] described it, 'adjustment is felt to be easy and successful to begin; then follows a "crisis" in which one feels less well adjusted' (p. 51). After the honeymoon stage, people encounter frustration, resulting from overload by a new culture, called the 'culture shock' stage. Then, during the 'recovery' stage, they develop skills to deal with difficulties from adjusting themselves to a new culture and, finally, become more integrated into the new culture [37, 38]. Applying this model to our

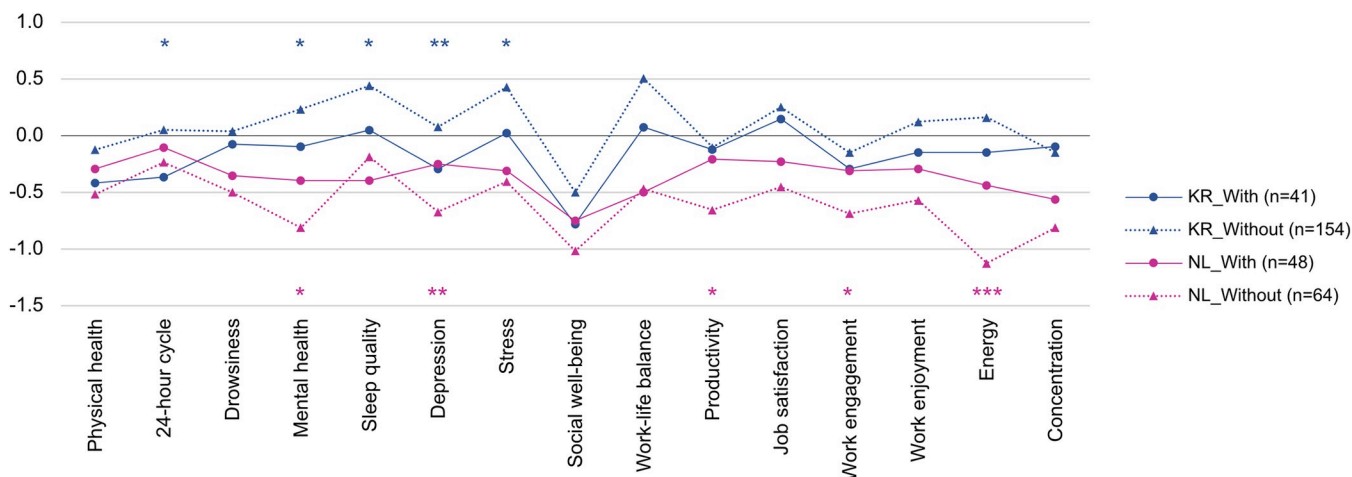

**Fig 4. Comparison of changes in health and productivity between people with experience of WFH before COVID-19 and those without the experience.** Blue represents the t-test results between the two groups (with and without prior WFH experience) of KR respondents, and pink represents the results of NL respondents. *$p<0.05$, **$p<0.01$, ***$p<0.001$.

results, KR can be said to have entered the honeymoon stage of WFH, somewhat excited about the benefits of the new work type. On the other hand, people in NL might currently be going through the time between stages of frustration and recovery, where they must overcome the difficulties of adaptation rather than enjoy the excitement and curiosity. To confirm this assumption more precisely, we performed independent-samples t-tests on groups with and without prior WFH experience for KR and NL, respectively (see Fig 4). Full results of the t-tests can be found in the S3.4 and S3.5 Tables of S3 Appendix.

In the KR group, we found that in terms of mental health, telecommuters without prior experience had more positive WFH experiences during the pandemic than those with prior experience. In contrast, NL respondents with prior experience tended to adapt better to WFH, notably in terms of productivity and mental health, than those with no previous experience. There were no significant differences between the two Korean groups in terms of productivity-related indicators, an interesting point. This may suggest that the appraisal of WFH by Korean telecommuters mostly depended on their emotional satisfaction rather than work efficiency or job engagement. This notion seems in line with the assumption above that Koreans may stay in the 'honeymoon' stage where they enjoy the thrill and excitement of a new work type.

Based on the results in Fig 2, it may seem that Koreans were better at adapting to the WFH measures during the pandemic than those in NL. However, it is possible that Koreans are currently going through the honeymoon stage of the new form of work, and when this period ends, the real difficulties and obstacles of WFH may emerge, one after another. While recent surveys in KR also keep reporting high levels of satisfaction with WFH among Koreans [39–41], concluding that WFH was already well-established in KR based only on these findings may be premature. Rather, it would be more desirable to understand what must be considered to establish and stably operate WFH by looking at the example of NL, which seems already to have passed the 'honeymoon' phase of WFH and is currently undergoing a tough adjustment period.

## 3.4. Frequency and compulsoriness of WFH (H6)

As noted in the Introduction, we assumed that the measures related to staying at home or WFH, differently implemented in KR and NL during COVID-19, may have affected WFH

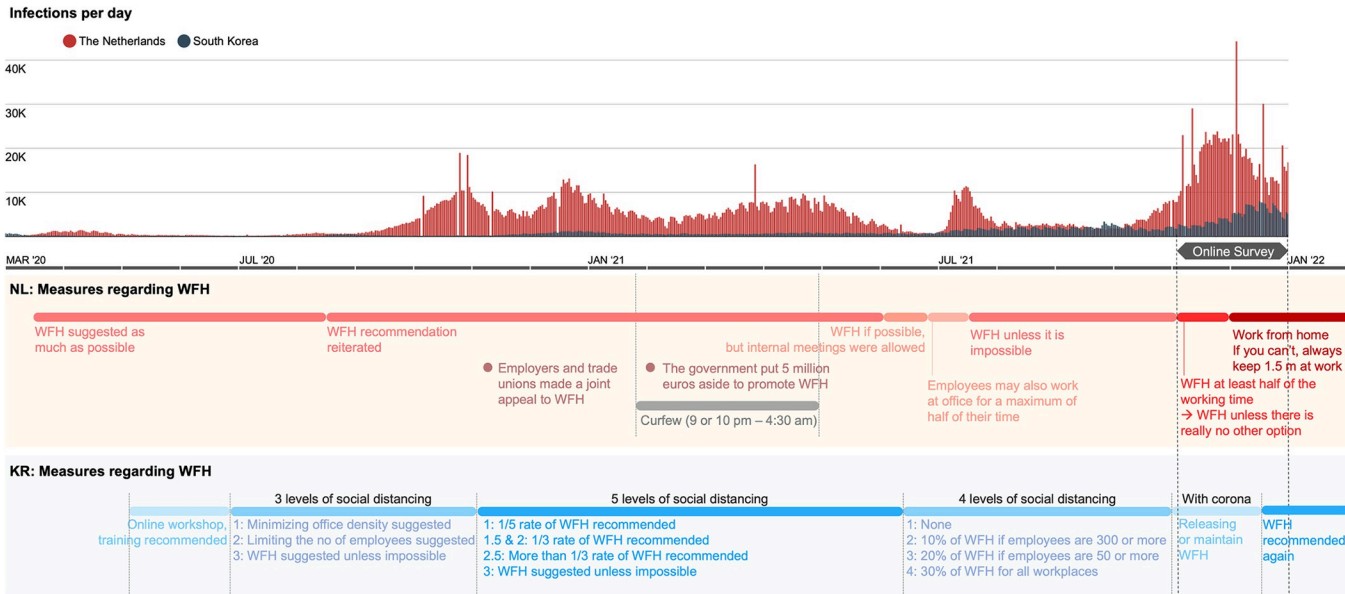

**Fig 5.** (a) Infections per day in NL and KR; (b) COVID-19 WFH-related measures in NL; (c) COVID-19 WFH-related measures in KR. Adapted from (a) Corona Board [42]; (b) European Commission Joint Research Centre [43]; (c) Korea Disease Control and Prevention Agency [44].

satisfaction and work productivity. Fig 5 depicts WFH-related COVID-19 measures in both countries from March 2020 to December 2021.

Neither country has ever mandated WFH for private companies except for special industries, such as contact professions. The Dutch government issued its first recommendation in March 2020 that as much teleworking be done as possible. This was temporarily relaxed between June and July 2021, allowing employees to work in the office for up to half of their working hours. However, as the number of infections rose, WFH was recommended again in July 2021, unless impossible. In November 2021, when the number of infections began to increase sharply, the measures were strengthened by recommending WFH for more than half of the working hours. Then, the rules for WFH were tightened again in late November, to 'Work from home. If this is impossible, stay 1.5 meters apart at work'.

On the other hand, Korea issued the first recommendation in May 2020, about online workshops or training and flexible work arrangements. More specific measures for WFH appeared at the end of June 2020 when a three-level social-distancing scheme was introduced. Although the rules were differentiated according to the number of infected people, a specific WFH ratio was not yet presented. In November 2020, when the three-level scheme was changed to the five-level rules, the specific WFH ratio first appeared. In the case of the 2.5 level, the highest level actually applied, more than one-third of the WFH ratio was recommended, which was the highest WFH ratio the Korean government suggested during the entire pandemic period. In particular, during the period when the online survey of this study was conducted (November–December 2021), the 'With Corona' policy (meaning the restoration of daily life) was initiated, and the measures for WFH and video conferences were relaxed.

Although both countries have only issued recommendations for WFH, NL seems to have taken a stronger stance on WFH than KR. The Dutch government used the terms 'as much as possible' or 'unless impossible' when recommending WFH, while the Korean government mentioned keeping WFH at a 'proper' ratio, such as 1/5 or 1/3. The strongest measures of both

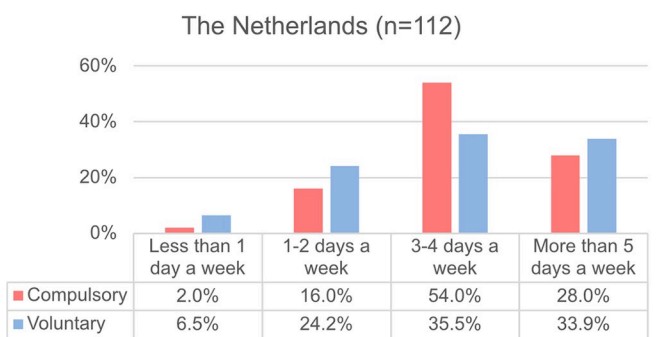
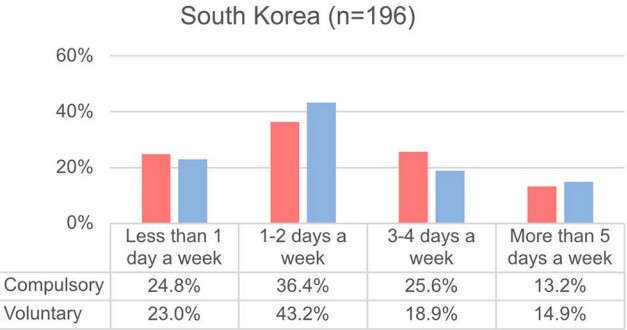

**Fig 6. Frequency of WFH of compulsory and voluntary groups in NL and KR.**

countries were also different. The Netherlands recommended more than half of the working hours, and Korea recommended one-third of the WFH rate.

We assumed that these differentiated measures of both countries might have affected an individual's frequency of WFH in the two countries and whether WFH was conducted voluntarily or compulsorily under the company's individual policy. As Fig 6 shows, we studied the relationship between the frequency and compulsoriness of WFH in each country. In the case of people working from home compulsorily in NL, more than 50% of them worked from home 3–4 days a week, and 28.0% worked from home 5 days or more. On the other hand, in KR, the rate of WFH 1–2 days a week was the highest among the compulsory group, and the rate of 1 or 3–4 days a week was around 25%, respectively. This result indicates that individuals in NL required to work from home had to remain at home longer than those in KR.

Considering this result, together with the result in Fig 2 that KR respondents scored higher on all outcome variables than NL participants, we hypothesised that when WFH is not implemented based on workers' willingness, the frequency of WFH would have a greater impact on their work productivity and health than when WFH is voluntary. For example, if two individuals work from home 4 days per week, and one person does so voluntarily while the other's situation is compulsory, the impact of this frequency of WFH on each individual's productivity and health may vary. To test this hypothesis, we divided the participants into two groups based on the compulsory nature of WFH, then divided each group into two subgroups based on the frequency of WFH (i.e. whether they worked from home more than 3 days a week). We compared the two subgroups for each of the compulsory and voluntary groups, in terms of changes in health and productivity, using independent-samples t-tests (see Fig 7). Full results of the t-tests can be found in S3.6 and S3.7 Tables of S3 Appendix.

As Fig 7 shows, the group of '2 days or less' scored higher than the group of '3 days or more' for almost every variable, in both compulsory and voluntary groups. However, more significant differences appeared according to the frequency of WFH in the compulsory group than in the voluntary group, indicating that the frequency of WFH is likely to make a difference in health and productivity when people must work from home. This result demonstrates the importance of the frequency of WFH for maintaining workers' health and well-being, as well as their productivity, especially in the pandemic situation where people were forced to stay at home, regardless of their will. In other words, for those whose work from home is compulsory, giving them at least the autonomy to determine the frequency or flexibility of WFH would be desirable. This result also emphasises the importance of introducing a hybrid working model, which offers employees more flexibility for where and when to work, as a way to adapt WFH to the post-corona era more stably. A recent BBC News article presented that 55% of U.S. employees prefer a mixture of home and office working [45].

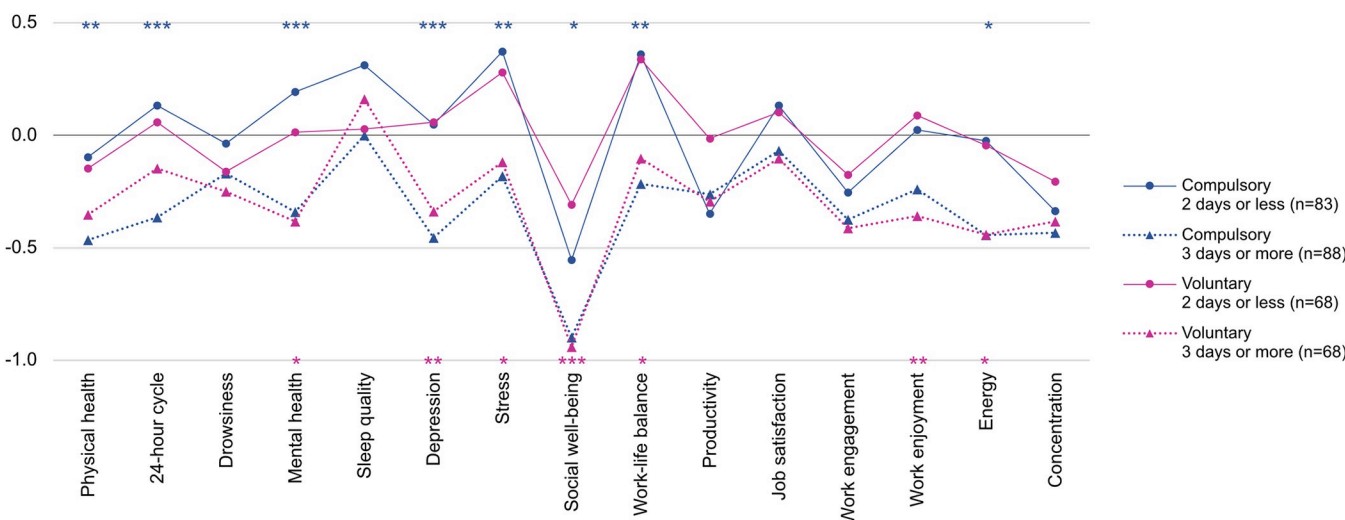

**Fig 7. Comparison of changes in health and productivity between compulsory group and voluntary group.** Blue represents the t-test results between the two groups ('2 days or less' and '3 days or more') of compulsory group, and pink represents the results of voluntary group. *$p<0.05$, **$p<0.01$, ***$p<0.001$.

## 3.5. Required interaction with colleagues or superiors during work (H7)

Finally, we discuss the results of this study from the perspective of the difference in workplace culture between the two countries. This would be a very complicated but important research topic that requires in-depth and extensive exploration of the multilayered and complex influence of national or organisational culture on WFH itself and telecommuters. As this is outside the subject of this study, drawing conclusions regarding this issue from our data may be unreasonable. However, to hint at future research topics, we present what can be very fragmentary research results on the impact of the different workplace cultures in KR and NL on telecommuters during the pandemic.

According to Hofstede-Insights [17], the most dominant contrast between KR and NL is the collectivism of KR and the individualism of NL. A collectivistic work culture prioritises the requirements of the 'group' above the needs of the 'individual'. Hence, a collectivist work culture stresses the role of a team member more than that of an individual worker and requires more intensive interaction between group members. While the collectivistic mindset fosters loyalty and collaboration among employees, it also creates unnecessary pressure by causing workers to feel additional guilt towards the team for their mistakes or poor performance [46].

As noted, we questioned how much interaction with colleagues or superiors is required during work, to investigate the differences in work culture between the two countries. We then performed an independent-samples t-test comparing KR and NL on the degree of interaction demanded. Consequently, the KR respondents ($M$ = 3.37, $SD$ = 0.884) compared to the NL participants ($M$ = 2.29, $SD$ = 0.875) demonstrated a significantly higher level of interaction required during work ($t$ = -3.680, $p = <$ .001), suggesting that this interaction demand level may be an indicator that reflects KR's collectivistic work culture. The regression analysis in Fig 2 shows that this variable had a substantial negative effect on KR respondents exclusively. These findings suggest that Korean employees, who are rooted in a collectivistic culture, may be more prone to experiencing additional stress, due to excessive concern about how group members see or judge them.

Himawan in 2022 [15] argued that WFH in Asian nations with a high level of collectivism could cause a sense of social isolation and disconnection among teleworkers. However, our

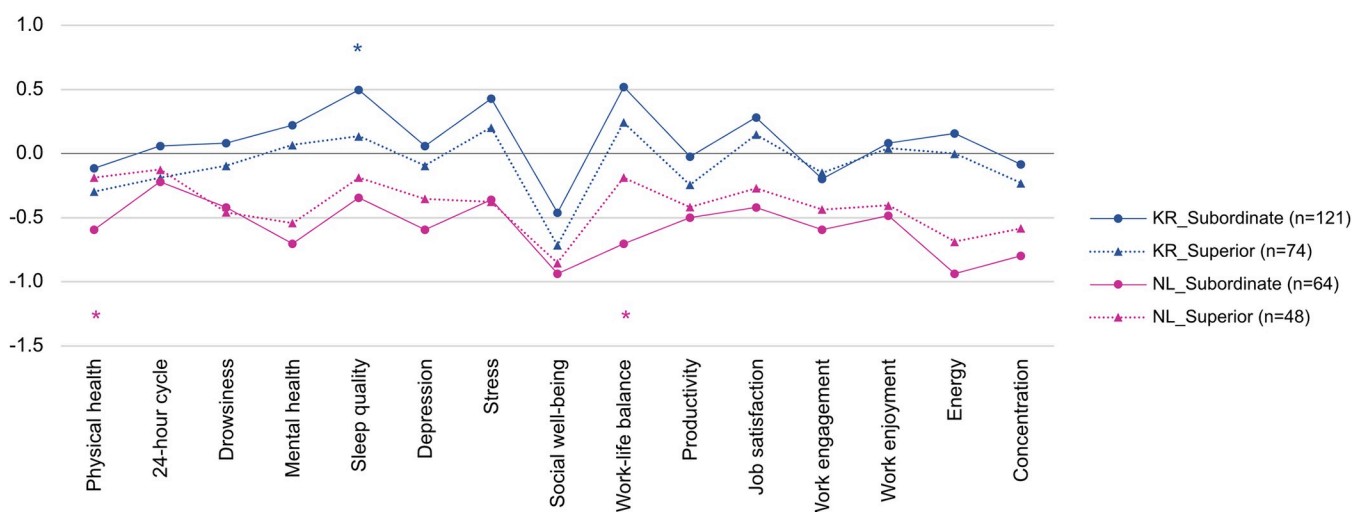

**Fig 8. Comparison of changes in health and productivity between subordinate group and superior group in both countries.** Blue represents the t-test results between the two groups (subordinate and superior) of KR respondents, and pink represents the results of NL respondents. *$p<0.05$, **$p<0.01$, ***$p<0.001$.

results confirmed that Koreans in a highly collectivist society experienced WFH during COVID-19 more positively than NL respondents, even in terms of social well-being. This may be because Korean workers have more favourably accepted WFH, which entails physical separation from groups that require constant connection and engagement. This coincides with the fact that 33.5% of respondents in a survey by the Seoul Foundation of Women and Family cited reducing stress due to unnecessary relationships at the office as the biggest advantage of WFH [47].

Another cultural dimension is power distance. Although we anticipated a strong influence by job position for KR, where the difference in power by position is particularly large (i.e. high power distance), no significant findings were seen for the KR respondents (see Fig 2). Instead, we found that only NL participants tended to report more positive changes in physical health and work-life balance as they advanced in their job positions. To explore this issue in more detail, we compared the mean values of outcome variables according to job positions (i.e. whether superior or subordinate) for KR and NL, respectively, as Fig 8 shows. Full results of the t-tests can be found in S3.8 and S3.9 Tables of S3 Appendix.

Interestingly, in the case of KR, the mean value of the subordinate group was greater than that of the superior group for all variables except 'work engagement'. In contrast, the superior group of the NL respondents scored higher than or similar to the subordinate group. In other words, while KR telecommuters in higher job positions tended to experience deterioration in health and productivity during the pandemic, NL telecommuters in superior job positions were likely to experience WFH better than those in subordinate job positions. This result should not prompt the conclusion that WFH can be experienced more positively by subordinate workers than superior workers in a high-power-distance society, but it does suggest that differences in work culture regarding power distance between workers can influence the relationship between job position and WFH satisfaction.

## 4. Conclusions

This study presents the results of an online survey conducted in the Netherlands and South Korea, to compare the two countries on the WFH experiences of telecommuters during

COVID-19 and the variables affecting changes in their health and productivity during working from home. The study produced the following findings:

1. Participants in KR had higher scores than those in NL in almost every aspect of changes in health and productivity and ratings for home workspace. However, concluding that WFH has been well established in KR based only on the current favourable outcomes of health and productivity may be premature, and preparing for difficulties and obstacles of WFH that may appear in the near future by learning from NL would be beneficial.

2. Respondents' WFH environment satisfaction, in terms of stress relief and concentration, turned out to be a set of significant predictors for changes in health and productivity during the pandemic in both countries. A negative effect of attachment to home on concentration in the KR group demonstrates the importance of separating work and family for comfortable living and a productive WFH environment.

3. Among the NL respondents, people living in houses had higher average scores than those in apartments, in almost every aspect of health and productivity changes and ratings for home workspaces. In particular, the average score for attachment to neighbours was lower among participants in apartments than in houses.

4. We found that the frequency of WFH is more important, especially for those who must work from home. Thus, allowing people who are forced to work from home to determine the frequency or flexibility of WFH would be desirable. This also shows the importance of introducing a hybrid working model as a way to prepare for the post-coronavirus era.

5. KR, which has a strong collectivistic work culture, tended to demand more interaction among team members during work than NL participants encountered. For those living in a collectivistic work culture, WFH may have made their experience of teleworking more positive by providing physical distance from the high demands of interaction between team members at work. We also found that how equal power distribution is in the working environment tends to differentiate the experience of WFH according to job position. In particular, in a high-power-distance society, subordinate workers may be more satisfied with WFH than superior workers because WFH allows subordinate telecommuters the physical distance from superiors who have more power than themselves.

Contrary to our expectations that the prevalence of WFH in NL prior to the COVID-19 pandemic would have made the people in NL more resilient to pandemic-induced WFH, this study found that a variety of sociocultural and personal factors had complex effects on health and productivity changes. This suggests that strategies for establishing a desirable WFH setting should take into account diverse underlying contexts, experiences and cultures, rather than looking for just a few universally applicable factors. In addition, as we used the U-curve of culture shock to illustrate the process of adjusting to WFH in a society, the establishment of WFH adaptation strategies will need a rigorous longitudinal investigation to validate this premise more comprehensively.

Our work has some limitations. First, although the 'differences' in health and productivity before and during COVID-19 were used as the main dependent variable of this study, we could not conduct a longitudinal study to measure them. Instead, survey participants had to not only assess their current state but also recall their health and productivity status before COVID-19 began, which may have caused information bias. Second, although we tried to randomly recruit participants using various social media in both countries, we were not able to guarantee broad coverage of the population. Third, we only analysed the participants' ratings for their home workspaces as variables representing the current conditions of home offices

and did not include the objective features of housing (e.g. house size, number of rooms, presence of windows, presence of art objects or plants). Our recent research investigated how these features of the built environment relate to the occupants' health and productivity while WFH and affect the perception of home workspaces [13]. We expect that the present study, along with our recent study focusing on how architectural factors of a WFH setting affect telecommuters' satisfaction, well-being and productivity, would help to enhance our understanding of how improving the design of indoor environments could support WFH better, as a way to prepare for the post-coronavirus era.

## Supporting information

**S1 Appendix. Measuring and coding variables.**
(DOCX)

**S2 Appendix. Regression analyses.**
(DOCX)

**S3 Appendix. Independent-samples t-tests.**
(DOCX)

**S4 Appendix. The U-curve of culture shock.** Adapted from Lysgaard [36].
(TIF)

**S5 Appendix. Raw data of this study.**
(XLSX)

## Author Contributions

**Conceptualization:** So Yeon Park, Rachel Lee, Caroline Newton.

**Data curation:** So Yeon Park.

**Formal analysis:** So Yeon Park.

**Funding acquisition:** So Yeon Park.

**Investigation:** So Yeon Park, Gisung Han.

**Methodology:** So Yeon Park, Rachel Lee, Caroline Newton.

**Project administration:** So Yeon Park.

**Resources:** So Yeon Park.

**Software:** So Yeon Park.

**Supervision:** Rachel Lee, Caroline Newton.

**Validation:** So Yeon Park.

**Visualization:** So Yeon Park.

**Writing – original draft:** So Yeon Park.

**Writing – review & editing:** So Yeon Park, Rachel Lee, Caroline Newton.

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
