## [Decision Letter · Decision Letter 0]

15 Aug 2023

PONE-D-23-15875How are people coping with working from home during the COVID-19 pandemic? : Experiences from the Netherlands and South KoreaPLOS ONE

Dear Dr. Park,

Thank you for submitting your manuscript to PLOS ONE. After careful consideration, we feel that it has merit but does not fully meet PLOS ONE’s publication criteria as it currently stands. Therefore, we invite you to submit a revised version of the manuscript that addresses the points raised during the review process.

We look forward to receiving your revised manuscript.

Kind regards,

Amin Yazdani, PhD

Academic Editor

PLOS ONE

“This work was supported by the National Research Foundation of Korea (NRF) grant funded by the Korea government (MEST) [No. NRF-2021R1A6A3A03040072].”

Reviewers' comments:

Reviewer's Responses to Questions

**Comments to the Author**

1. Is the manuscript technically sound, and do the data support the conclusions?

Reviewer #1: Yes

Reviewer #2: Yes

2. Has the statistical analysis been performed appropriately and rigorously? 

Reviewer #1: Yes

Reviewer #2: Yes

3. Have the authors made all data underlying the findings in their manuscript fully available?

Reviewer #1: Yes

Reviewer #2: Yes

4. Is the manuscript presented in an intelligible fashion and written in standard English?

Reviewer #1: Yes

Reviewer #2: Yes

5. Review Comments to the Author

Reviewer #1: This article examines the health and productivity changes in employees working from home before and during the COVID-19 pandemic in Korea and Netherlands. The authors found better health and productivity scores in Koreans compared to the workers in Netherlands, and the scores were also affected by different aspects of work- and home-related factors. The findings can contribute to the growing literature on work-from-home. The following suggestions can help the authors improve the manuscript.

Abstract

- Line 20: Use “pandemic” instead of “epidemic”

Introduction

- Line 55: Use “pandemic” instead of “epidemic”

- Line 94: This hypothesis is unclear. Are the authors hypothesizing that the predictor variables within the “Satisfaction” component (presented in Figure 1) may significantly affect the health and productivity outcomes, and that the significant predictors might be different for both KR and NL?

Methods

- Line 146: Instead of beginning a sentence with the in-text citation number, I suggest including the author’s name first and then the citation number. For example, Pang et al. [21]

- Line 169: Use “pandemic” instead of “epidemic”.

- Line 191: Use “performed” instead of “perform”.

- Line 196: Figure 1 does not include the following outcome variables: “24-hour cycle”, ”Drowsiness”, “Sleep quality”, “Depression”, “Stress”. Those variables were included in Figure 2; therefore, I suggest presenting them in Figure 1 too for clarity.

- Lines 198-200: Were participants excluded from the study if they answered Yes or No to those 4 questions? Please specify.

- Lines 204-205: The numbers presented here adds up to 306; however, authors stated that 307 participants were included in the study (line 201). Please clarify the discrepancy.

Results and Discussion

- Line 249: The use of “WFH experience” is unclear. Based on Figure 2, “WFH experience” is one of the predictor variables. Are the authors referring to the health and productivity outcomes instead of WFH experience? Please clarify.

- Lines 258-264: I suggest including a reference to Figure 2 or the supplementary tables for some of the findings presented here as it will be helpful for the reader.

- Line 285: The use of “partially significant” is unclear. Are authors stating that the housing variable has a significant effect on only some of the health and productivity outcomes? Please clarify. Also, based on Figure 2, the housing variable has a significant effect only for Korean workers, and not for the workers in Netherlands; therefore, the use of “both countries” is unclear.

- Line 302-304: This sentence is unclear. Based on Figure 3, the scores of the house group are significantly higher than the apartment group only for 7 of the variables. For the remaining variables, there were no statistically significant differences between the house and apartment groups. Please clarify.

- Line 314: This sentence is unclear because according to Figure 2, “attachment to neighbours” affected social well-being in Korean workers. Please clarify.

- Please use a period instead of comma as the decimal separator in Figures 3, 4, 6, 7, and 8.

- Line 427-428: When authors use the term “satisfaction”, are they referring to “job satisfaction” (which is an outcome variable in Fig 2) or “satisfaction with WFH space” (which is a predictor variable in Fig 2)? Please clarify.

- Line 479: Instead of beginning a sentence with the in-text citation number, I suggest including the author’s name first and then the citation number.

Conclusions

- Line 517: Does the word “satisfaction” refer to the “satisfaction with WFH space” (based on Fig 2) or are the authors referring to the health and productivity outcomes? Please clarify.

- Line 536-537: This sentence is unclear. Please clarify.

Reviewer #2: This is a very nice manuscript that addresses the changes in health and productivity that occurred as a result of working from home during the pandemic. The work is novel as it compares the changes in outcomes and various factors associated with changes in both the Netherlands and South Korea. The analyses are very comprehensive and the interpretations of the findings are appropriate and well grounded in the literature. The tables and figures are well constructed and accurate. The appendix contains the details of the statistical models and the summary results are accurately presented in each of the graphs.

Some comments on the manuscript are:

- Ln 53-54 suggest a reference for the development index

- Ln 113 it would be appropriate to have a paragraph at the beginning of the methods section that describes the study design and population.

- In the methods section there should be clarification of how questions were asked to obtain information on variables before and during the pandemic (e.g., Ln 126 the discussion of before/after values) – it should be clarified this is a cross sectional study that collects this information at one point in time.

- The section on predictive variables in interesting but is more of a review and does not get to the details of how the various variables are actually measured. It would be nice if, after reading the section on the Research Variables that the reader would be able to figure how each of the variables were measured and scored. Figure 2 is helpful. It seems that the predictive variables that don’t have the way they are coded specified were assessed using a 5 point likert scale.

- Ln 197 It is odd that half of the respondents were excluded from the analysis. It would be nice to be more specific with the exclusion criteria and indicate the number excluded due to the specific reason. In terms of question 4 – are you Korean for the NL questionnaire – why is this an exclusion?

- Ln 329 this is a minor point but just wondering why past experience with WFH is coded 0?

-

6. PLOS authors have the option to publish the peer review history of their article (what does this mean?). If published, this will include your full peer review and any attached files.

Reviewer #1: No

Reviewer #2: **Yes: **Philip Bigelow

While revising your submission, please upload your figure files to the Preflight Analysis and Conversion Engine (PACE) digital diagnostic tool, https://pacev2.apexcovantage.com/. PACE helps ensure that figures meet PLOS requirements. To use PACE, you must first register as a user. Registration is free. Then, login and navigate to the UPLOAD tab, where you will find detailed instructions on how to use the tool. If you encounter any issues or have any questions when using PACE, please email PLOS at <a href="mailto:figures@plos.org

---

## [Author Response · Author response to Decision Letter 0]

22 Oct 2023

Response to Reviewer #1’s comments:

This article examines the health and productivity changes in employees working from home before and during the COVID-19 pandemic in Korea and Netherlands. The authors found better health and productivity scores in Koreans compared to the workers in Netherlands, and the scores were also affected by different aspects of work- and home-related factors. The findings can contribute to the growing literature on work-from-home. The following suggestions can help the authors improve the manuscript.

** Answer: Thank you for your constructive feedback on our manuscript. We greatly appreciate the time and effort you invested in reviewing our work and are encouraged by your recognition of its potential contribution to the literature on working from home. The valuable comments and suggestions have allowed us to significantly improve our manuscript. In the following, we include point-by-point responses to the comments. The related revision in the manuscript is highlighted in red. The line numbers indicated in the answer are based on the ‘Simple Markup’ version of the revised manuscript. We do hope our responses/revisions fully resolve the concerns that you have raised below.

1) Line 20: Use “pandemic” instead of “epidemic”

2) Line 55: Use “pandemic” instead of “epidemic”

3) Line 169: Use “pandemic” instead of “epidemic”

** Answer: We have replaced “epidemic” with “pandemic” as suggested (Line 22, 57, and 206). Thank you for pointing that out.

4) Line 94: This hypothesis is unclear. Are the authors hypothesizing that the predictor variables within the “Satisfaction” component (presented in Figure 1) may significantly affect the health and productivity outcomes, and that the significant predictors might be different for both KR and NL?

** Answer: You are correct. We hypothesized that the predictor variables within the satisfaction component that have a significant impact on the variables may vary between KR and NL participants. This has been clarified in the text (Line 96-98).

5) Line 146: Instead of beginning a sentence with the in-text citation number, I suggest including the author’s name first and then the citation number. For example, Pang et al. [21]

6) Line 479: Instead of beginning a sentence with the in-text citation number, I suggest including the author’s name first and then the citation number.

** Answer: We have adjusted the in-text citation format to include the author’s name first, as suggested (Line 183 and 493). Thanks for pointing that out.

7) Line 191: Use “performed” instead of “perform”.

** Answer: The word “perform” has been changed to “performed” for proper tense consistency (Line 122).

8) Line 196: Figure 1 does not include the following outcome variables: “24-hour cycle”, ”Drowsiness”, “Sleep quality”, “Depression”, “Stress”. Those variables were included in Figure 2; therefore, I suggest presenting them in Figure 1 too for clarity.

** Answer: We have now included the mentioned outcome variables in Figure 1 to maintain clarity and consistency. Thank you for your careful pointing out.

9) Lines 198-200: Were participants excluded from the study if they answered Yes or No to those 4 questions? Please specify.

** Answer: We further elaborated in the revised manuscript on how we selected participants based on their answers to each of the four questions (Line 133-134).

10) Lines 204-205: The numbers presented here adds up to 306; however, authors stated that 307 participants were included in the study (line 201). Please clarify the discrepancy.

** Answer: We apologize for the mistake. We now realize that there were two respondents who said they did not want to disclose their gender, not one, and have corrected the number accordingly (Line 139).

11) Line 249: The use of “WFH experience” is unclear. Based on Figure 2, “WFH experience” is one of the predictor variables. Are the authors referring to the health and productivity outcomes instead of WFH experience? Please clarify.

** Answer: You are right. We meant the health and productivity outcomes. This has been clarified (Line 261-262).

12) Lines 258-264: I suggest including a reference to Figure 2 or the supplementary tables for some of the findings presented here as it will be helpful for the reader.

** Answer: We have now referenced Figure 2 on line 268 to make it easier for readers to navigate (Line 272).

13) Line 285: The use of “partially significant” is unclear. Are authors stating that the housing variable has a significant effect on only some of the health and productivity outcomes? Please clarify. Also, based on Figure 2, the housing variable has a significant effect only for Korean workers, and not for the workers in Netherlands; therefore, the use of “both countries” is unclear.

** Answer: Thanks for pointing that out. By “partially significant”, we meant that the housing variable was significant for some outcomes but not all. The notation “both countries” was an error that occurred as we were evolving the draft paper. We have corrected this error and changed the wording in this paragraph entirely for clarity (Line 297-303). 

14) Line 302-304: This sentence is unclear. Based on Figure 3, the scores of the house group are significantly higher than the apartment group only for 7 of the variables. For the remaining variables, there were no statistically significant differences between the house and apartment groups. Please clarify.

** Answer: Thank you for your thoughtful point. While the differences were not statistically significant, we were referring to the differences between the apartment and house groups shown in the “mean scores”. We have revised the text to more clearly convey this point (Line 316-318).

15) Line 314: This sentence is unclear because according to Figure 2, “attachment to neighbours” affected social well-being in Korean workers. Please clarify.

** Answer: Thanks for pointing that out. This was another error that occurred during the revision process of the first draft paper. “Attachment to neighbours” was found to have a positive effect on various outcome variables for the NL participants, but a negative effect on social well-being for the KR participants. We have clarified this in the text (Line 327-329).

16) Please use a period instead of comma as the decimal separator in Figures 3, 4, 6, 7, and 8.

** Answer: We have updated Figures 3, 4, 6, 7, and 8 to use a period as the decimal separator, as suggested.

17) Line 427-428: When authors use the term “satisfaction”, are they referring to “job satisfaction” (which is an outcome variable in Fig 2) or “satisfaction with WFH space” (which is a predictor variable in Fig 2)? Please clarify.

18) Line 517: Does the word “satisfaction” refer to the “satisfaction with WFH space” (based on Fig 2) or are the authors referring to the health and productivity outcomes? Please clarify.

** Answer: We meant general satisfaction with WFH in terms of health and productivity. We have clarified our use of the term in response to your point (Line 442 and 532).

19) Line 536-537: This sentence is unclear. Please clarify.

** Answer: This sentence has been revised for clarity (Line 549-551).

We are grateful for your constructive feedback. It has certainly enhanced the quality and clarity of our manuscript. We hope the revisions meet your expectations and look forward to further feedback.

Response to Reviewer #2’s comments: 

This is a very nice manuscript that addresses the changes in health and productivity that occurred as a result of working from home during the pandemic. The work is novel as it compares the changes in outcomes and various factors associated with changes in both the Netherlands and South Korea. The analyses are very comprehensive and the interpretations of the findings are appropriate and well grounded in the literature. The tables and figures are well constructed and accurate. The appendix contains the details of the statistical models and the summary results are accurately presented in each of the graphs.

** Answer: Thank you very much for your encouraging feedback on our manuscript. We are delighted to hear that you found the study valuable, especially in its comparative approach between the Netherlands and South Korea, and its thoroughness in analysis. The valuable comments and suggestions have allowed us to significantly improve our manuscript. In the following, we include point-by-point responses to the comments. The related revision in the manuscript is highlighted in red. The line numbers indicated in the answer are based on the ‘Simple Markup’ version of the revised manuscript. We do hope our responses/revisions fully resolve the concerns that you have raised below. 

1) Ln 53-54 suggest a reference for the development index

** Answer: Thank you for pointing that out. We have added a reference for the development index and corrected the incorrect year (Line 56 and 594-596).

2) Ln 113 it would be appropriate to have a paragraph at the beginning of the methods section that describes the study design and population.

** Answer: Thank you for your valid point. We have moved the existing section “2.2. Online survey” to section “2.1” and re-titled it “2.1. Study design and sample”. This rearrangement ensures that the methodology and recruitment of participants for this study are discussed at the beginning of Chapter 2 (Line 115-143).

3) In the methods section there should be clarification of how questions were asked to obtain information on variables before and during the pandemic (e.g., Ln 126 the discussion of before/after values) – it should be clarified this is a cross sectional study that collects this information at one point in time.

** Answer: We appreciate for your important point. To provide more specific information about the survey questions to measure the outcome variables, we have provided an example of how the two questions were actually constructed to calculate the difference in one of the outcome variables (i.e., physical health) (Line 161-164).

4) The section on predictive variables in interesting but is more of a review and does not get to the details of how the various variables are actually measured. It would be nice if, after reading the section on the Research Variables that the reader would be able to figure how each of the variables were measured and scored. Figure 2 is helpful. It seems that the predictive variables that don’t have the way they are coded specified were assessed using a 5 point likert scale.

** Answer: To provide more information on how the variables were measured, we have presented how we coded all outcome and predictive variables in this study as supporting information (i.e., S1 Measuring and coding variables) in the revised manuscript. We added a reference to this material at the end of “2.2. Research variables” (Line 217-218). 

5) Ln 197 It is odd that half of the respondents were excluded from the analysis. It would be nice to be more specific with the exclusion criteria and indicate the number excluded due to the specific reason. 

** Answer: Participation in this survey was limited to those who (1) worked from home in NL or KR during the pandemic, (2) had worked in a space other than their home before COVID-19, (3) worked at a desk for a significant amount of time, and (4) were non-Koreans living in NL. In the original text, we presented the four questions we used in the online survey to determine these four criteria, but for further clarification, we elaborated in the revised manuscript on how we selected participants based on their answers to each of the four questions (Line 133-134).

Unfortunately, we were unable to determine how many participants were excluded based on what specific criteria, as the survey was designed to immediately discard the responses of excluded participants. However, given that multiple accesses were allowed to this pre-survey, it is possible that this number does not represent the actual number of excluded participants.

6) In terms of question 4 – are you Korean for the NL questionnaire – why is this an exclusion?

** Answer: Thank you for your valid point. We recognized that Koreans living in the Netherlands may have been influenced by both Korean and Dutch culture and policies, which may have affected the consistency and interpretability of our findings. We intentionally excluded Koreans living in the Netherlands from the survey to avoid diluting the clarity of our main objective, which is to compare working from home in Korea and the Netherlands. We have added lines 123-125 to the text to make this clear.

7) Ln 329 this is a minor point but just wondering why past experience with WFH is coded 0?

** Answer: For the nominal variable, we have randomly coded one value as 0 and the other as 1. The choice was arbitrary, with no specific reason behind it.

We greatly appreciate the time and effort that you have dedicated to improving our manuscript. We believe that the modifications we have made in light of the comments have strengthened the paper, and we hope that the revised version meets the expectations. We look forward to hearing your feedback.

---

## [Decision Letter · Decision Letter 1]

15 Mar 2024

How are people coping with working from home during the COVID-19 pandemic? : Experiences from the Netherlands and South Korea

PONE-D-23-15875R1

Dear So Yeon,

We’re pleased to inform you that your manuscript has been judged scientifically suitable for publication and will be formally accepted for publication once it meets all outstanding technical requirements.

Kind regards,

Xiaoqiang ‘Jack’ Kong

Academic Editor

PLOS ONE

Additional Editor Comments (optional):

Reviewers' comments:

Reviewer's Responses to Questions

**Comments to the Author**

1. If the authors have adequately addressed your comments raised in a previous round of review and you feel that this manuscript is now acceptable for publication, you may indicate that here to bypass the “Comments to the Author” section, enter your conflict of interest statement in the “Confidential to Editor” section, and submit your "Accept" recommendation.

Reviewer #1: All comments have been addressed

2. Is the manuscript technically sound, and do the data support the conclusions?

Reviewer #1: Yes

3. Has the statistical analysis been performed appropriately and rigorously? 

Reviewer #1: Yes

4. Have the authors made all data underlying the findings in their manuscript fully available?

Reviewer #1: Yes

5. Is the manuscript presented in an intelligible fashion and written in standard English?

Reviewer #1: Yes

6. Review Comments to the Author

Reviewer #1: (No Response)

7. PLOS authors have the option to publish the peer review history of their article (what does this mean?). If published, this will include your full peer review and any attached files.

Reviewer #1: No

---

## [Editor Report · Acceptance letter]

8 Apr 2024

PONE-D-23-15875R1 

PLOS ONE

Dear Dr. Park, 

I'm pleased to inform you that your manuscript has been deemed suitable for publication in PLOS ONE. Congratulations! Your manuscript is now being handed over to our production team.

Kind regards, 

on behalf of

Dr. Xiaoqiang ‘Jack’ Kong 

Academic Editor

PLOS ONE